# Continuous flash suppression of neural responses and population orientation coding in macaque V1

Cai-Xia Chen[1†], Xin Wang[1†], Dan-Qing Jiang[1], Shi-Ming Tang[2,3]*, Cong Yu[4,5]*

[1]School of Psychological and Cognitive Sciences, Peking University, Beijing, China; [2]School of Life Sciences, Peking University, Beijing, China; [3]IDG-McGovern Institute for Brain Research, Peking University, Beijing, China; [4]Department of Psychology and Behavioral Sciences, Zhejiang University, Hangzhou, China; [5]Zhejiang Key Laboratory of Neurocognitive Development and Mental Health, Zhejiang University, Hangzhou, China

*For correspondence:
tangshm@pku.edu.cn (S-MT);
yu_cong@zju.edu.cn (CY)

[†]These authors contributed equally to this work

Competing interest: The authors declare that no competing interests exist.

## eLife Assessment

This **important** study shows that orientation tuning of V1 neurons is suppressed during a continuous flash suppression paradigm, especially in neurons with binocular receptive fields. These findings, made using cutting-edge imaging techniques, **convincingly** implicate early visual processing in continuous flash suppression, in agreement with previous studies suggesting reduced effective contrast of such stimuli in V1.

**Abstract** Continuous flash suppression (CFS), in which a dynamic masker presented to one eye suppresses awareness of a stimulus in the other eye, is widely used to study visual subconsciousness. Although some studies report preserved high-level processing under CFS, these effects have been increasingly questioned and may partly reflect residual low-level feature processing. A key unresolved issue is how strongly neuronal responses in V1, where inputs from the two eyes first converge, are affected by CFS, and how much the remaining signals can support downstream processing. Here, we used two-photon calcium imaging to record large populations of V1 neurons in awake, fixating macaques while presenting grating stimuli under CFS. CFS strongly suppressed V1 orientation responses in an ocular-dominance-dependent manner, nearly abolishing responses in neurons preferring the masker eye or both eyes, and significantly reducing responses in neurons preferring the grating eye. Modeling analyses further indicated that V1 population activity under CFS may still support coarse orientation classification but not accurate stimulus reconstruction. These results suggest that CFS substantially degrades orientation information in V1. The residual signals may support limited low-level processing but are likely insufficient for downstream higher-level visual and cognitive tasks.

## Introduction

When a target stimulus is presented to one eye and a flickering Mondrian-like masker to the other eye, the target can be rendered invisible for an extended period (*Tsuchiya and Koch, 2005*). This paradigm, known as continuous flash suppression (CFS), has been widely used to investigate subconscious visual processing (e.g. *Yang et al., 2014*; *Moors et al., 2017*; *Pournaghdali and Schwartz, 2020*). Among the most intriguing findings are the subconscious high-level visual and cognitive functions under the influence of CFS (e.g. *Fang and He, 2005*; *Almeida et al., 2008*; *Adams et al., 2010*;

*Zabelina et al., 2013*; *Tettamanti et al., 2017*). For example, as reported, priming effects are evident when the target and the invisible primer are categorically (*Almeida et al., 2008*) or semantically (*Zabelina et al., 2013*) consistent. However, many of these observations have been questioned by more recent studies, with at least some of the high-level effects being attributed to low-level feature processing (e.g. *Hesselmann and Malach, 2011*; *Sakuraba et al., 2012*; *Moors et al., 2017*; *Pournaghdali and Schwartz, 2020*).

A critical issue in this debate is the impact of CFS on V1 neuronal activity. CFS has been hypothesized to arise from mechanisms similar to those in binocular rivalry (*Tsuchiya and Koch, 2005*; *Yang et al., 2014*; *Moors et al., 2017*), which likely suppress V1 responses through interocular inhibition. Only the surviving stimulus information would then be relayed to downstream areas for potential subconscious higher-level visual and cognitive processing (*Jiang et al., 2007*; *Adams et al., 2010*; *Almeida et al., 2010*). Importantly, if V1 activity is suppressed to a sufficient degree, the low-level stimulus information carried by the remaining V1 responses may not suffice to sustain high-level processing of more complex stimuli defined by those low-level features.

Two prominent fMRI studies have examined the impact of CFS on V1 activity (*Watanabe et al., 2011*; *Yuval-Greenberg and Heeger, 2013*), but they reached different conclusions. *Watanabe et al., 2011*, compared monocular CFS masking (stimulus visible) and dichoptic CFS masking (stimulus invisible) and reported that V1 BOLD responses were largely insensitive to stimulus visibility when attention was carefully controlled. However, using similar experimental design, *Yuval-Greenberg and Heeger, 2013*, observed reduced BOLD responses in V1 under dichoptic masking, suggesting that V1 activity changed with stimulus visibility. They attributed the difference in results between the two studies mainly to differences in the number of trials and thus the statistical power (~250 trials per condition vs. ~90 trials per condition). Nevertheless, these studies contrasted monocular and dichoptic masking conditions to equate stimulus input while manipulating perceptual visibility, which were not designed to quantify the full effect of CFS on stimulus-evoked V1 responses, i.e., the difference in BOLD signals between binocular masking and stimulus alone conditions. In contrast, original psychophysical studies (*Tsuchiya and Koch, 2005*; *Tsuchiya et al., 2006*) demonstrated CFS masking by contrasting the visibility of the target stimulus with and without the presence of dichoptic mask. Consequently, the impact of CFS on V1 activity should be larger than what has been reported by *Yuval-Greenberg and Heeger, 2013*.

Neurons in V1 exhibit various degrees of ocular dominance (OD) (*Hubel and Wiesel, 1962*), which influences each neuron's binocular combination of monocular visual inputs from two eyes (*Kato et al., 1981*; *Mitchell et al., 2023*; *Zhang et al., 2024*). In the present study, we used a with-or-without-dichoptic-masker design similar to those used in original psychophysical studies and examined the extent to which V1 neuronal responses were affected by CFS and how neurons preferring the target eye, masker eye, or both eyes were differently impacted. Using a customized two-photon imaging setup for awake macaques (*Li et al., 2017*), we sampled large neural populations at cellular resolution and calculated OD for each individual neuron. This approach enabled us to investigate the potentially differential impacts of CFS on the responses of V1 neurons with varying ocular preferences, as well as to apply machine learning tools to understand the impacts of CFS on V1 orientation coding at the population level.

## Results

We used two-photon calcium imaging to record responses of V1 superficial neurons from two awake, fixating macaques, each with two response fields of view (FOVs, 850×850 μm$^2$) (*Figure 1A*). During the initial recording, the stimulus was a binocular 0.45-contrast square-wave grating varying at twelve orientations and two spatial frequencies (3 and 6 cpd) (*Figure 1B*). A total of 3564 neurons were identified through image processing, including 3004 (84.29%) orientation-tuned neurons that were included in the following data analyses.

The same grating stimulus was then presented monocularly (*Figure 1B*) to each eye to characterize individual neurons' eye preferences. Each neuron's ocular dominance index (ODI) was calculated as ODI = $(R_i - R_c)/(R_i + R_c)$, where $R_i$ and $R_c$ were the neuron's peak responses to ipsilateral and contralateral stimulations, respectively. Neurons with an ODI at –1 or +1 would exclusively prefer the contralateral or ipsilateral eye, while neurons with an ODI at 0 would prefer both eyes equally. Consistent with previous findings (*Hubel and Wiesel, 1962*; *Horton and Hocking, 1996*; *Livingstone, 1996*;

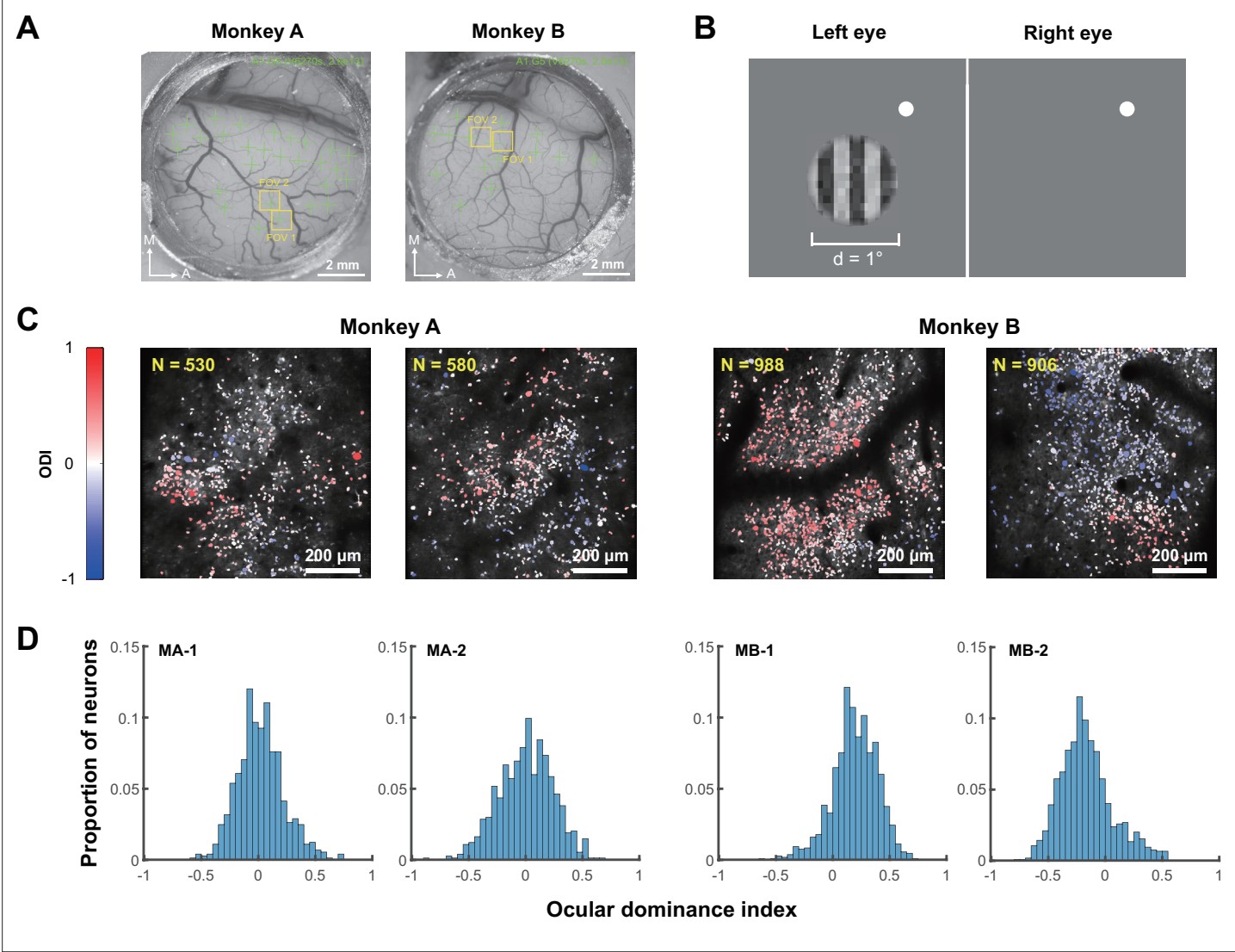

**Figure 1.** Two-photon imaging and ocular dominance (OD) mapping. (**A**) Optical windows for imaging of two macaques. Green crosses indicate the regions for viral vector injections, and yellow boxes indicate the fields of view (FOVs) chosen for imaging. (**B**) Stimuli used for OD mapping. A circular-windowed square-wave grating was presented monocularly to each eye, respectively, to probe each neuron's ocular dominance index (ODI). (**C**) OD functional maps of each FOV at single-neuron resolution showing OD clusters. (**D**) Frequency distributions of individual neurons' OD indices in each FOV.

*Zhang et al., 2024*), neurons with similar eye preferences clustered together (*Figure 1C*). The ODI followed unimodal distributions (*Figure 1D*), in which the majority of neurons were binocular, showing comparable preferences for either eye. Only a small portion of neurons were monocular, being more responsive to the ipsilateral or contralateral eye.

In a third and last step, the grating stimulus and the flashing noise masker were presented dichoptically to evaluate the impact of CFS on neurons' orientation responses (*Figure 2A*). The results are summarized as population orientation tuning functions under the baseline no-CFS condition and the CFS condition following the procedure in *Busse et al., 2009*. Specifically, neurons with similar orientation preferences were binned (bin width = 15°) relative to the target orientation for a total of 12 bins, and the resultant population orientation tuning functions based on the mean responses of these bins (*Figure 2C*) were fitted with a Gaussian function. Compared to the baseline population orientation tuning functions, those under the influence of CFS displayed profound reductions in orientation response. The amplitude decreased by 84.18% in Monkey A and 60.78% in Monkey B on the basis of Gaussian fitting, while the slope decreased by 91.31% in Monkey A and 71.50% in Monkey B

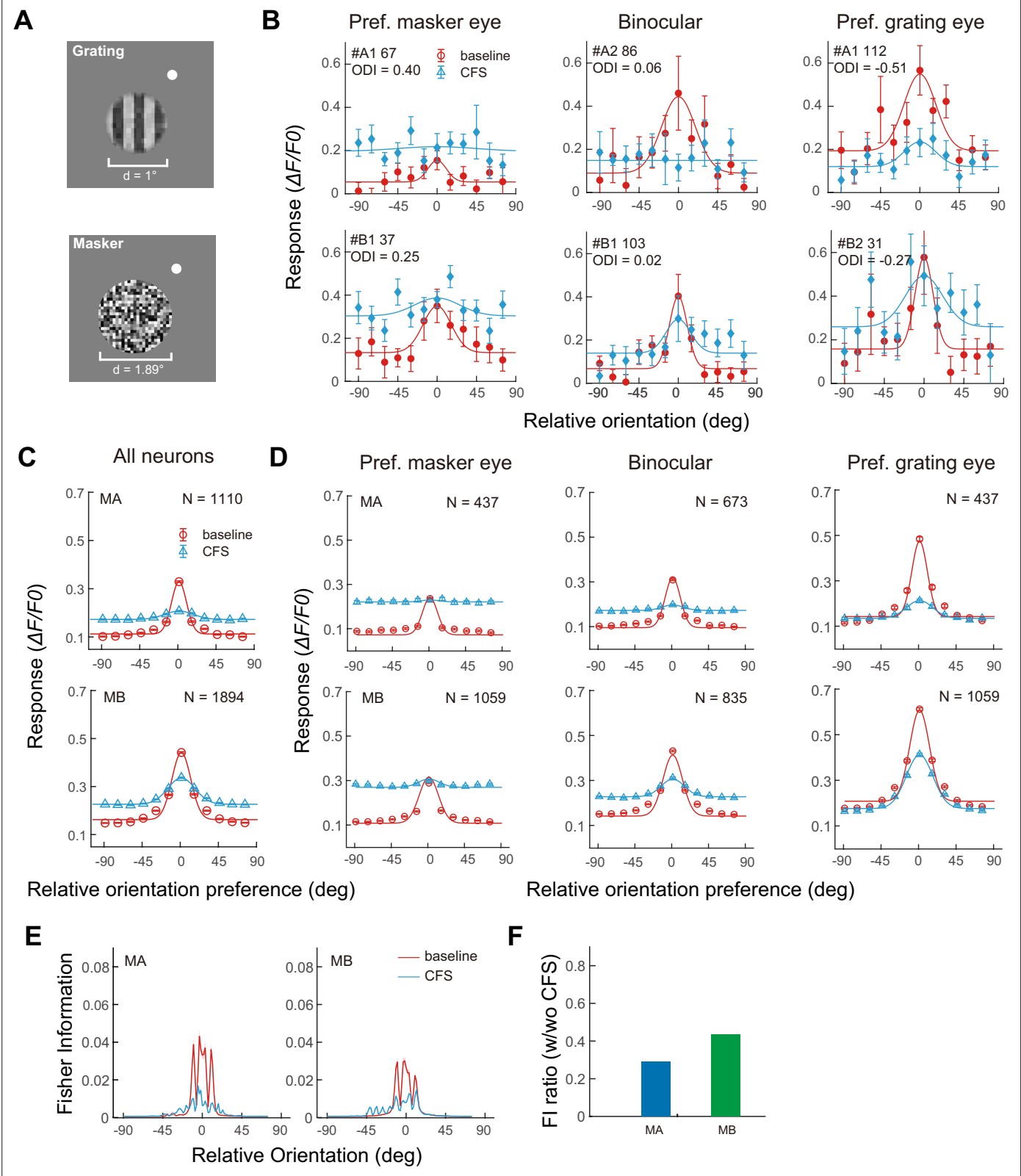

**Figure 2.** The impacts of continuous flash suppression (CFS) on population orientation tuning in two macaques. (**A**) Stimuli used in the CFS experiment for one macaque. The grating target was presented to one eye, which was dichoptically masked by a circular flashing masker presented to the other eye. The white dot was the fixation point. (**B**) Exemplar baseline and CFS orientation tuning functions for neurons with different eye preferences based on Gaussian fitting. Error bars indicate ± SE across trials (*n* = 12 trials per condition for all FOVs, except *n* = 10 trials for MB2). (**C**) Population orientation

*Figure 2 continued on next page*

*Figure 2 continued*

tuning functions of all neurons without CFS as the baseline and with CFS based on Gaussian fitting. Data from two FOVs of each monkey were pooled due to highly consistent results. Error bars represent ±1 SE across neurons. (**D**) Population orientation tuning functions of subgroups of neurons with different eye preferences without and with CFS. Data from two FOVs of each monkey were pooled. Curves are fitting results using an ocular-dominance-dependent gain control model. Error bars represent ±1 SE across neurons. (**E**) The impacts of CFS on Fisher information. Fisher information is plotted as a function of relative orientation to the neuron's preferred orientation without and with CFS. Shaded areas denote ±1 SE across neurons. (**F**) The ratio of baseline/CFS Fisher information within 15° of neurons' preferred orientations. Data from two FOVs of each monkey were pooled due to highly consistent results.

(*Figure 2B*). These results support the conclusion of *Yuval-Greenberg and Heeger, 2013*, that V1 activity is degraded by CFS, 'akin' to a loss in the contrast-to-noise ratio of neural activity.

Furthermore, neurons were divided into three groups according to their ODIs, and the impacts of CFS on their respective orientation responses were examined: neurons preferring the grating eye (ODI >0.2 or <–0.2, depending on whether the grating stimulation was ipsilateral or contralateral), binocular neurons (–0.2 ≤ ODI ≤ 0.2), and neurons preferring the masker eye (ODI <–0.2 or >0.2 relative to the grating eye). Compared to the baseline condition, the orientation tuning of neurons preferring the masker eye was completely abolished by CFS (*Figure 2B and D*, left), leading to flat-tened tuning curves with unmeasurable amplitudes and bandwidths. The orientation tuning of binocular neurons was either nearly completely abolished (Monkey A) or substantially abolished (Monkey B) (*Figure 2B and D*, middle). There were 85.68% and 68.32% decreases in amplitude, and 92.64% and 77.07% decreases in slope, for Monkeys A and B, respectively. The orientation tuning of neurons preferring the grating eye was the least but still substantially affected (*Figure 2B and D*, right), with respective 77.78% and 41.75% decreases in amplitude and 85.23% and 57.56% decreases in slope for two monkeys.

To quantify the loss of V1 population orientation encoding due to CFS, we compared the Fisher information (*Averbeck and Lee, 2006*) under both baseline and CFS conditions. Here, Fisher infor-mation serves as a statistical measure that reflects how much information the responses of neurons can provide about the grating orientation. Specifically, it indicates the sensitivity of neural responses to small changes in orientation, in that higher values signify greater precision in encoding orienta-tion information. As illustrated in *Figure 2E*, Fisher information was reduced by CFS primarily for orientations deviating by less than 15° from the neurons' preferred orientations. The average Fisher information for stimuli within this 15° range decreased to 29.1% and 43.4% of the baseline values in two macaques, respectively (*Figure 2F*), indicating the detrimental impact of CFS on the ability of V1 populations to accurately encode and represent orientation information, especially for orientations closely aligned with neuronal preferences.

## An OD-dependent gain control model

We developed an OD-dependent gain control model to account for the impact of CFS on V1 popula-tion orientation tuning. The model development followed two steps.

### Step I. Population orientation tuning functions before CFS

The population orientation tuning functions due to monocular stimulation exhibited different ampli-tudes among OD groups (*Figure 2D*, red curves), which could be simulated with *Equation 1*, an OD-weighted Gaussian basis function:

$$\boldsymbol{R} = \boldsymbol{w}^s \times \left[ A \times exp^{-\left(\frac{\theta}{\sigma}\right)^2} + B \right] \quad (1)$$

where parameters $A$, $\sigma$, and $B$ corresponded to the amplitude, standard deviation, and minimal response of the Gaussian basis function, respectively, and $\theta$ represented the preferred orientation of a bin of neurons relative to the actual orientation of the grating stimulus. The weight parameter $w$ was the mean of linearly transformed ODIs of neurons in a neuronal group, which equated to (ODI + 1)/2 or 1 – (ODI + 1)/2, depending on contralateral or ipsilateral eye grating stimulation, and ranged from 0 to 1. Thus, a smaller $w$ would indicate a higher preference for the eye seeing the grating, and a larger

$w$ would indicate a higher preference for the unstimulated eye (or the eye seeing the flashing masker under CFS). The $w$ equaled 0.33, 0.50, and 0.67 in Monkey A, and 0.32, 0.5, and 0.68 in Monkey B, for the grating eye-preferring group, binocular group, and masker eye-preferring group, respectively. The exponent $s$ represented a nonlinear transformation.

*Equation 1* fitted the baseline data well (*Figure 2D*, red curves), resulting in goodness-of-fit ($R^2$) values at 0.94 and 0.95 for the two monkeys, respectively. This indicated that the equation very well captured the OD-dependent population orientation tuning characteristics of V1 neurons with monocular stimulation before CFS.

## Step II. The impacts of CFS

In step II, the model introduced several binocular combination factors to account for population orientation tuning functions under CFS.

To account for the OD-dependent changes of orientation tuning bandwidths under CFS, a $w$-dependent inhibition factor $w^t$ was introduced, which scaled the $\sigma$ of the tuning functions, changing the monocular tunings $R$ into $R'$:

$$R' = w^s \times \left[ A \times exp^{-\left( \frac{\theta}{\sigma \times w^t} \right)^2} + B \right] \qquad (2)$$

This allowed different groups of neurons to exhibit various degrees of broadening of orientation tuning functions, capturing the pattern in which neurons preferring the eye seeing the grating displayed a sharper population orientation tuning curve under CFS than those preferring the eye seeing the masker.

Previous studies have shown that binocular neuronal responses can be modeled by incorporating interocular suppression and summation processes (*Kato et al., 1981*; *Dougherty et al., 2019*; *Zhang et al., 2024*). Therefore, $R'$ was further normalized by the neural response to the flashing masker to simulate interocular suppression, which was the first component of *Equation 3*. Additionally, the neural response to the flashing masker, which was the second component of *Equation 3*, was summed to simulate binocular summation. The final neural response under CFS was:

$$R_{CFS} = \frac{R'}{aN^k} + bN^m \qquad (3)$$

where $N$ was the empirical neural response to the monocularly presented flashing masker stimulation, $a$ and $b$ were scaling parameters, and $k$ and $m$ were nonlinearity parameters. The interocular normalization by masker response led to amplitude reduction of population orientation tuning functions for different groups of neurons, while the binocular summation with masker response elevated the minimal responses of tuning functions to their corresponding heights.

During the step II model fitting, the values of parameters $A$, $\sigma$, and $s$ were inherited from the monocular tuning fits derived from *Equation 1* and served as fixed inputs, while the parameters $a$, $k$, $b$, $m$, and $t$ were optimized. The model captured the CFS modulation on population orientation tuning curves well, with $R^2$=0.99 and 0.98 for Monkeys A and B, respectively (*Figure 2D*, red curves).

## Orientation classification and reconstruction under CFS

What is the impact of CFS-induced suppression on V1 orientation decoding? To answer this question, which is crucial for understanding subconscious processing under CFS, we trained linear decoders to classify stimulus orientations in our experiments, as well as transformer models to reconstruct the stimulus images. Here, orientation classification was analogous to coarse orientation discrimination, and image reconstruction was analogous to orientation recognition, both suggesting the upper bounds of performance assuming an ideal observer.

For orientation classification, we trained an all-pair multiclass support vector machine (SVM) classifier to discriminate 12 orientations based on trial-by-trial population neural responses from all trials (*Allwein et al., 2000*). Decoders for different FOVs, ipsilateral/contralateral target presentations, and baseline vs. CFS conditions were trained separately. Under the baseline condition, the decoders achieved mean classification accuracies of 89.5 ± 2.0% and 91.5 ± 2.1% across ipsilateral

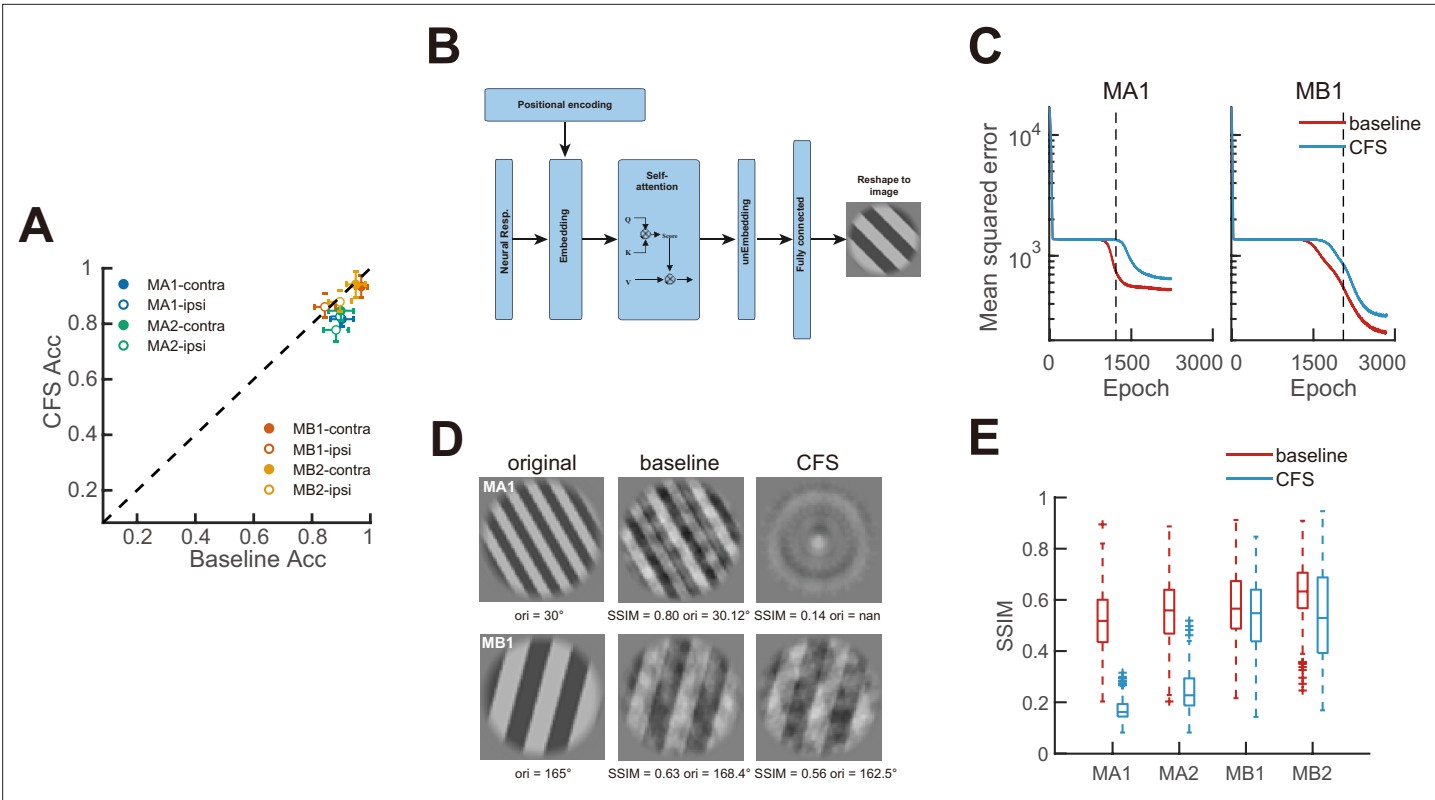

**Figure 3.** Decoding consequences of continuous flash suppression (CFS) revealed by machine learning. (**A**) Multiway orientation classification accuracies under CFS vs. baseline conditions obtained using support vector machine (SVM) decoders. Each datum represents results from a contralateral or ipsilateral grating condition with a specific FOV averaged across 10-fold cross-validations. Error bars denote 95% confidence intervals across cross-validation folds. (**B**) A diagram of the transformer model for stimulus image reconstruction. (**C**) Exemplar learning curves of transformer models under baseline and CFS conditions from two FOVs. The vertical dashed line indicates the epoch at which the baseline model reaches 75% of its total loss decrease between the two learning plateaus estimated using a sigmoid fit. (**D**) Illustrations of corresponding reconstructed stimulus images on the basis of learning curves in C. (**E**) Box plots of structural similarity index (SSIM) scores between the original and reconstructed images with baseline and CFS transformers. Within an FOV, results from contralateral eye and ipsilateral eye conditions are combined. Box summarizes the combined samples within an FOV (576 samples per FOV except 480 for MB2). Box plots show the median (center line), interquartile range (box), whiskers extending to 1.5× the interquartile range, and outliers (data points beyond the whiskers).

and contralateral eye conditions in Monkeys A and B, respectively, in contrast to a chance level of 8.3% (1 out of 12). Under CFS, decoding accuracy slightly decreased in Monkey A (81.7 ± 1.9%) but remained stable in Monkey B (90.4 ± 2.1%, *Figure 3A*). These results suggest that under CFS, there is still sufficient information for coarse orientation discrimination, even for Monkey A whose V1 neuronal responses were substantially suppressed.

Next, we trained transformer models to reconstruct the grating images on the basis of corresponding neuronal responses under baseline and CFS conditions. The motivation for this part of the modeling work was the assumption that high-level tasks would be difficult to carry out if the basic stimulus features forming more complex patterns were not intact. Our transformer model contained an architecture that integrated embedding, self-attention, and unembedding modules, as well as a fully connected feedforward layer (*Figure 3B*). The model inputs were the responses of all neurons within an FOV to the grating stimulus (ipsilateral and contralateral presentations of the same stimulus were modeled separately), and the model output was the reconstructed grating image. During the training process, the model typically reached two successive learning plateaus, where the validation loss temporarily stagnated (*Figure 3C*). Moreover, the validation loss decreased more rapidly when training on the baseline neural response data compared to the CFS data. To compare the differences, we identified the epoch at which the validation loss of the baseline model reached 75% of its total decrease between the two plateaus using a sigmoid fit, and then we retrained both the baseline and CFS models up to this epoch.

The retrained baseline models reconstructed the grating stimuli significantly better than the CFS models in Monkey A, but this discrepancy was less pronounced in Monkey B (*Figure 3D*), consistent with the neuronal data that Monkey A exhibited substantially more CFS suppression than Monkey B in terms of population orientation tuning and Fisher information (*Figure 2*). We used a structural similarity index (SSIM) (*Brunet et al., 2012*) to quantify the reconstruction performances. Across the grating-presenting ipsilateral and contralateral eyes, the baseline models reconstructed the grating with median SSIMs of 0.52 and 0.61 for the two FOVs of Monkey A, and 0.57 and 0.63 for the two FOVs of Monkey B, respectively, while the corresponding SSIMs for the CFS models were 0.16 and 0.19 for Monkey A, and 0.55 and 0.53 for Monkey B (*Figure 3E*).

To estimate the impact of CFS-induced V1 suppression on downstream processing, we also recorded neuronal responses from two V2 FOVs in Monkey A (FOVs V2-1 and V2-2). As anticipated, V2 neurons were binocular, with over 90% of them showing ODIs within the range of –0.2 to 0.2 (*Figure 4A*). Similar to V1 results from the same monkey, CFS on average reduced the amplitudes of the population orientation tuning functions by 80.05% and the slopes by 89.44% (*Figure 4B*). It also reduced the Fisher information to 33.1% of the baseline value (*Figure 4C*). Furthermore, we applied the same orientation classification and image reconstruction procedures to the V2 data. For orientation classification, the SVM decoders achieved near-perfect performance in classifying 12 orientations under both baseline and CFS conditions, with classification accuracies exceeding 94% across all cases (*Figure 4D*). In the image reconstruction task, the baseline model outperformed the CFS model. Specifically, the baseline transformer models reconstructed the stimulus images with the median SSIM values of 0.61 and 0.53 for the two V2 FOVs, respectively, which dropped to 0.42 and 0.18 in the CFS models (*Figure 4E*), implying poorer or failed reconstruction of stimulus images.

## Discussion

Our study demonstrates that CFS severely compromises orientation information in V1 neurons in an OD-dependent manner. Orientation information carried by neurons preferring the masker eye or both eyes is completely or nearly completely abolished, while information carried by those preferring the grating eye is partially retained. Downstream, orientation information in V2 neurons is also substantially weakened. Linear decoding and transformer models further suggest that CFS-compromised orientation information may still support coarse orientation discrimination, but is much less likely to support orientation recognition when the suppression is sufficiently strong, as in Monkey A. Suppression in Monkey B might also have been stronger if the grating contrast had been reduced to 0.1–0.3 (*Watanabe et al., 2011*; *Yuval-Greenberg and Heeger, 2013*; *Alais et al., 2024*).

CFS-compromised V1 orientation information may explain the unconscious orientation processing observed in human CFS studies. The adaptation aftereffect under CFS is reduced compared to the visible condition but not entirely abolished (*Kanai et al., 2006*; *Bahrami et al., 2008*), likely a result of the degraded orientation information surviving CFS. For the same reason, the priming effect also decreases in trials in which the stimulus is rendered invisible by CFS, compared to those in which the stimulus is visible or partially visible (*Koivisto and Grassini, 2018*), as the degraded stimulus information provides insufficient evidence for decision-making, resulting in a diminished priming effect (*Dehaene, 2011*; *Gomez et al., 2013*).

Our linear decoding and transformer results may help clarify the debate over whether visual processing under CFS still operates at the categorization level. Previous studies have reported preserved category information in suppressed targets, as demonstrated by tool-specific priming effects (*Almeida et al., 2008*; *Almeida et al., 2010*) and differential BOLD response patterns between tools and other object categories under CFS (*Hesselmann et al., 2011*; *Tettamanti et al., 2017*). However, these findings may instead reflect low-level feature differences between tools and other categories. For example, elongated objects, regardless of category, elicit similar priming effects (*Sakuraba et al., 2012*), and when tools are grouped by shape, only elongated tools can be discriminated from other object categories under CFS (*Fogelson et al., 2014*; *Ludwig et al., 2015*). Consistent with this view, *Hesselmann et al., 2018*, found that tool-specific priming under CFS does not reliably emerge under conditions of strong interocular suppression, suggesting that previously reported category effects may reflect access to low-level shape features rather than preserved category representations.

Moreover, a recent study measuring the contrast thresholds required to both break from and suppress CFS found that stimuli exhibited similar suppression strengths across various categories

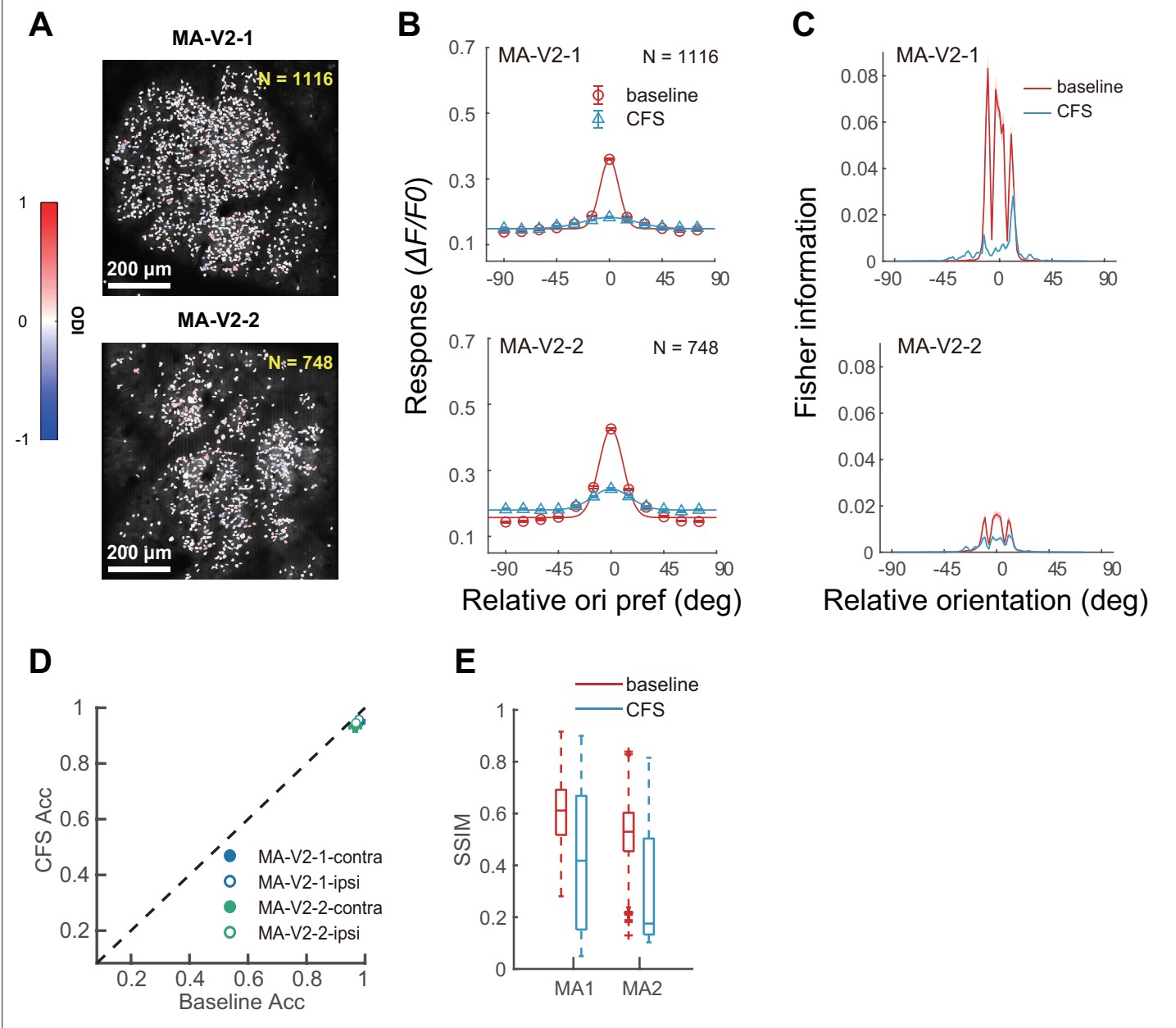

**Figure 4.** Effects of continuous flash suppression (CFS) on V2 orientation responses. (**A**) Ocular dominance (OD) maps of the two V2 FOVs of Monkey A (MA-V2-1 and MA-V2-2). (**B**) Population orientation tuning functions for all orientation-tuned neurons with baseline and CFS conditions. Curves represent the results of Gaussian fittings. Error bars represent ±1 SE across neurons. (**C**) Fisher information as a function of the relative orientation to the neuron's preferred orientation with baseline and CFS conditions. Shaded areas denote ±1 SE across neurons. Fisher information was lower in MA-V2-2 due to higher variations in the data. (**D**) Multiway orientation classification accuracies under CFS vs. baseline conditions using support vector machine (SVM) decoders. Each datum represents results from a contralateral or ipsilateral grating condition with one FOV, averaged across 10 fold cross-validations. Error bars denote 95% confidence intervals across cross-validation folds. (**E**) Box plots of structural similarity index (SSIM) scores between the original and reconstructed images with baseline and CFS transformers. Within an FOV, results from contralateral eye and ipsilateral eye conditions are combined. Box summarizes the combined samples within an FOV (672 samples for MA-V2-1, 480 for MA-V2-2). Box plots show the median (center line), interquartile range (box), whiskers extending to 1.5× the interquartile range, and outliers (data points beyond the whiskers).

(*Alais et al., 2024*). According to our results, when suppression is strong enough to prevent reliable stimulus reconstruction, as in Monkey A (*Figure 3C*), the orientation information that survives under CFS may not accumulate to a level sufficient to resolve semantic category boundaries. Such categorization may require a more intact stimulus representation, even if the processing remains subconscious. By contrast, residual information may still contribute to category discrimination when category

differences map onto low-level shape dimensions such as elongation or orientation, because coarse orientation discrimination appears relatively robust under CFS suppression (*Figure 3A*).

A related issue is the dorsal-ventral CFS hypothesis, which proposes that CFS may disproportionately affect ventral visual processing while relatively sparing dorsal pathways involved in visuomotor functions, potentially allowing category- or action-related information to remain accessible under suppression (*Fang and He, 2005*). However, subsequent fMRI studies have not provided consistent support for this dissociation, instead reporting either stream-invariant awareness effects (*Hesselmann and Malach, 2011*; *Ludwig et al., 2015*; *Tettamanti et al., 2017*), residual signal in ventral rather than dorsal regions (*Hesselmann et al., 2011*; *Fogelson et al., 2014*), or evidence more consistent with preserved low-level feature information or partial visibility than with preserved dorsal processing (*Ludwig et al., 2015*). Although our study does not directly test dorsal-ventral dissociations, our V1 results provide an important constraint on what information downstream visual pathways could access under suppression. When CFS-induced interocular suppression is strong enough and stimuli reconstruction is markedly reduced, as in Monkey A, the information required for category-level or action-related processing may also be insufficient for stable high-level cortical representation.

Interocular suppression under CFS is known to vary substantially across individuals (*Yamashiro et al., 2014*; *Gayet and Stein, 2017*; *Blake et al., 2019*), which may contribute to the heterogeneity observed in the CFS literature. We also found that the strength of V1 response suppression under CFS differed between two monkeys, as reflected in population orientation tuning functions (*Figure 2C*), Fisher information (*Figure 2F*), and reconstruction performance by the transformer (*Figure 3E*). Several experimental factors may have contributed to the relatively weaker suppression observed in Monkey B. Because the monkeys viewed the stimuli passively, we could not determine the dominant eye for each monkey (instead, we switched the eyes and averaged the results), and the target was presented at relatively high contrast. Both factors are known to reduce the effectiveness of CFS suppression (*Yang et al., 2010*; *Yuval-Greenberg and Heeger, 2013*). In addition, the random-noise masker we used might not be as effective as Mondrian patterns (*Hesselmann et al., 2016*). Nevertheless, our main conclusion was consistent across animals: CFS substantially reduced V1 orientation responses, and in Monkey A, this suppression was strong enough to markedly impair stimulus reconstruction.

## Materials and methods

**Key resources table**

| Reagent type (species) or resource | Designation | Source or reference | Identifiers | Additional information |
|---|---|---|---|---|
| strain, strain background (*M. mulatta*) | Rhesus monkey | Beijing Prima Biotech | | http://www.primabio.com.cn/Default |
| recombinant DNA reagent | AAV1.hSyn.GCaMP5G | Penn Vector Core | V5072MI-R | |
| software, algorithm | MATLAB | MathWorks | RRID:SCR_001622 | R2022b |

## Monkey preparation

Monkey preparation was identical to procedures reported in previous studies (*Ju et al., 2021*; *Guan et al., 2021*; *Zhang et al., 2024*). Two male rhesus monkeys (*Macaca mulatta*, aged 5 and 6, respectively) underwent two sequential surgeries under general anesthesia and strictly sterile conditions. During the first surgery, a 20 mm diameter craniotomy was performed on the skull over V1. The dura was opened, and multiple tracks of 100–150 nil AAV1.hSynap.GCaMP5G.WPRE.SV40 (AV-1-PV2478, titer 2.37e13 (GC/ml), Penn Vector Core) were pressure-injected at a depth of ~350 μm at multiple locations. The dura was then sutured, the skull cap was re-attached with three titanium lugs and six screws, and the scalp was sutured. After the surgery, the animal was returned to the cage and treated with injectable antibiotics (Ceftriaxone sodium, Youcare Pharmaceutical Group, China) for 1 week. Postoperative analgesia was also administered. The second surgery was performed 45 days later. A T-shaped steel frame was installed for head stabilization, and an optical window was inserted onto the cortical surface. Data collection could start as early as 1 week later. More details about the preparation and surgical procedures can be found in *Li et al., 2017*. All experimental protocols were approved by the Peking University Animal Care and Use Committee (LSC-TangSM-3).

## Behavioral task

After a 10-day recovery period following the second surgery, monkeys were placed in a primate chair with head restraint. They were trained to hold fixation on a small white spot (0.2°) with eye positions monitored by an Eyelink-1000 eye tracker (SR Research) at a 1000 Hz sampling rate. During the experiment, trials with the eye position deviated 1.5° or more from the fixation before stimulus offset were discarded as trials containing saccades and repeated.

## Visual stimuli and experimental procedures

Visual stimuli were generated with a MATLAB-based Psychtoolbox-3 software (*Pelli and Zhang, 1991*) and presented on an ROG Swift PG278QR monitor (refresh rate = 120 Hz, resolution = 2560 × 1440 pixel, pixel size = 0.23 × 0.23 mm$^2$). The screen luminance was linearized by an 8-bit look-up table, and the mean luminance was 47 cd/m$^2$. The viewing distance was 60 cm.

A drifting square-wave grating (spatial frequency = 4 cpd, contrast = full, speed = 3 cycles/s, starting phase = 0°, size = 0.4° in diameter) was first used to determine the population receptive field (pRF) location, shape, and approximate size associated with a specific FOV. The same stimulus was also monocularly presented to confirm the V1 location as OD columns would appear. This fast process used a 4× objective lens mounted on the two-photon microscope and did not provide cell-specific information. The recorded V1 pRFs were centered at ~0.90° eccentricity in Monkey A and ~1.93° in Monkey B. V2 pRFs were centered at ~0.67° in Monkey A. All pRFs were approximately circular with a diameter of 0.9°.

The target stimulus used in the experiments was a 0.45-contrast circular-windowed square-wave grating. It drifted at 4 cycles/s in opposite directions perpendicular to the orientation with a starting phase of 0° and varied at 12 orientations (0° to 165° in 15° increments) and two spatial frequencies (3 and 6 cpd) trial by trial. The circular envelope had a diameter of 1°, which approximated the size of pRFs for recorded FOVs, with the edge blurred by a linear ramp starting at a radius of 0.38°. The flashing masker was a circular white noise pattern with a diameter of 1.89°, a contrast of 0.5, and a flickering rate of 10 Hz. The white noise consisted of randomly generated black and white blocks (0.07°×0.07° each). The target grating and the flashing masker were presented through a pair of NVIDIA 3D Vision 2 active shutter glasses. To mitigate the ghost image, a low-contrast (RMS contrast = 0.08) white noise was added to the grating. The width of the noise element was half of the bar width of the square grating, and the white noise was regenerated every frame.

Each block of trials consisted of four groups of stimuli: binocular, monocular, CFS, and flashing masker-only. In the binocular group, the grating was presented to both eyes simultaneously. The relevant data were only used to help identify regions of interest (ROIs) and orientation-tuned neurons along with data from other stimulus conditions. In the monocular group, the grating was monocularly presented to the contralateral or ipsilateral eye, which served as the baseline conditions without the influences of CFS. In the CFS group, the grating and flashing masker were presented dichoptically. In the flashing masker-only group, the flashing masker was presented monocularly to either eye. Each stimulus condition was repeated for 10–14 times, depending on the FOV (10 trials in FOV MB2, 14 trials in FOV MA-V2-2, 12 trials in the others). For conditions involving the grating, the trials were split for two opposite drifting directions. A block of trials contained 242 trials, two trials for each stimulus condition, with the order of stimulus conditions arranged in a pseudorandom manner. There were 5–7 blocks of trials with each FOV.

Each stimulus was presented for 1000 ms, followed by an interstimulus interval of 1500 ms, allowing sufficient time for the calcium signals to return to the baseline level (*Guan et al., 2020*). For each FOV, the recording was completed in a single session with 5–7 experiment blocks and lasted for 2–3 hr.

## Two-photon imaging

Two-photon imaging was performed using a FENTOSmart two-photon microscope (Femtonics), along with a Ti:sapphire laser (Mai Tai eHP, Spectra Physics). GCaMP5 was chosen as the indicator of calcium signals because the fluorescence activities it expresses are linearly proportional to neuronal spike activities within a wide range of firing rates from 10 to 150 Hz (*Li et al., 2017*). During imaging, a 16× objective lens (0.8 N.A., Nikon) with a resolution of 1.6 µm/pixel was used, along with a 1000 nm femtosecond laser. A fast resonant scanning mode (32 fps) was chosen to obtain continuous images of neuronal activity (8 fps after averaging every 4 frames). The strength of fluorescent signals (mean

luminance of a small area) was monitored and adjusted if necessary for the drift of fluorescent signals. Two response FOVs measuring 850×850 μm$^2$ in V1 were selected in both macaques, and two FOVs of the same size in V2 were selected in Macaque A.

## Imaging data analysis: initial screening of ROIs

Data were analyzed with customized MATLAB codes. A normalized cross-correlation-based translation algorithm was used to reduce motion artifacts (*Li et al., 2017*). Then the fluorescence changes were associated with corresponding visual stimuli through the time sequence information recorded by Neural Signal Processor (Cerebus system, Blackrock Microsystems). By subtracting the mean of the 4 frames before stimuli onset ($F0$) from the average of the 6th–9th frames after stimuli onset ($F$) across 5 or 6 repeated trials for the same stimulus condition (same orientation, spatial frequency, size, and drifting direction), the differential image ($\Delta F = F - F0$) was obtained.

For a specific FOV, the ROIs or possible cell bodies were decided through sequential analysis of 242 differential images in the order of CFS, monocular, binocular, and flashing masker-only conditions. CFS conditions consisted of 96 (2×2×12×2 = 96) differential images, with the grating presented to either eye (2), at two spatial frequencies (2), twelve orientations (12), and two motion directions (2). Monocular conditions were identical to the CFS conditions except that the flashing masker was absent. In the binocular conditions, gratings at two spatial frequencies (2), twelve orientations (12), and two motion directions (2) were binocularly presented, resulting in 48 differential images. The flashing masker-only conditions consisted of the flashing masker presented to either eye, resulting in two differential images.

The first differential image was filtered with a band-pass Gaussian filter (size = 2–10 pixels), and connected subsets of pixels (>25 pixels, which would exclude smaller vertical neuropils) with average pixel value >3 standard deviations of the mean brightness were selected as ROIs. Then the areas of these ROIs were set to mean brightness in the next differential image before the band-pass filtering and thresholding were performed. This measure gradually reduced the standard deviations of differential images and facilitated the detection of neurons with relatively low fluorescence responses. If a new ROI and an existing ROI from the previous differential image overlapped, the new ROI would be on its own if the overlapping area OA <1/4 ROI$_{new}$, discarded if 1/4 ROI$_{new}$ <OA < 3/4 ROI$_{new}$, and merged with the existing ROI if OA >3/4 ROI$_{new}$. The merges would help smooth the contours of the final ROIs. This process went on through all differential images twice to select ROIs. Finally, the roundness for each ROI was calculated as:

$$Roundness = \frac{\sqrt{4\pi \times A}}{P}$$

where $A$ was the ROI's area, and $P$ was the perimeter. Only ROIs with roundness larger than 0.9, which would exclude horizontal neuropils, were selected for further analysis.

## Imaging data analysis: orientation tuning and OD

The ratio of fluorescence change ($\Delta F/F0$) was calculated as a neuron's response to a specific stimulus condition. For a specific neuron's response to a specific stimulus condition, the $F0n$ of the $n$th trial was the average of 4 frames before stimulus onset (−500 to 0 ms), and $Fn$ was the average of the *5th-8th, 6th-9th, or 7th-10th* frames after stimulus onset, whichever was the greatest. $F0n$ was then averaged across 10 or 12 repeated trials to obtain the baseline $F0$ for all trials (to reduce noise in the calculation of responses), and $\Delta Fn/F0 = (Fn - F0)/F0$ was taken as the neuron's response to this stimulus at the $n$th trial.

Several steps were taken to determine whether a neuron was orientation-selective. For each monocular or binocular condition, the orientation and SF eliciting the maximal response were designated as the neuron's preferred SF and orientation. We then compared responses across all 12 orientations at the preferred SF by performing a non-parametric Friedman test to determine whether the neuron's responses at various orientations were significantly different from each other. To reduce Type I errors, the significance level was set at $\alpha=0.01$. Neurons that passed the Friedman test at least under one viewing condition were selected as orientation-tuned neurons.

The ODI was calculated to characterize each neuron's eye preference: ODI = $(R_i - R_c)/(R_i + R_c)$, where $R_i$ and $R_c$ were the neuron's peak responses at the best orientation and SF to ipsilateral and

contralateral monocular grating conditions, respectively. Neurons with an ODI of –1 or 1 would be completely contralateral or ipsilateral eye dominant, and those with an ODI of 0 would be equally dominant by both eyes.

## Population orientation tuning

For each neuron, neural responses at the preferred SF were selected for tuning analysis. To derive population orientation tuning curves under a specific condition, we categorized neurons into 12 orientation preference bins according to their preferred orientations (bin width = 15°). For each orientation presented, the responses of all orientation preference bins were reorganized according to the relative orientation preference. Subsequently, neuronal responses of the same relative orientation preference were averaged to generate the final population orientation tuning function. For CFS conditions, the selected SF and binning procedures were the same as their corresponding monocular conditions.

The population orientation tuning function was fitted with a Gaussian model with MATLAB's nonlinear least-squares function 'lsqnonlin':

$$R\left(\theta\right) = a \times 2^{-\left(\frac{\theta - \theta_0}{\sigma}\right)^2} + b$$

where $R(\theta)$ was the response at orientation $\theta$, free parameters $a$, $\theta_0$, $\sigma$, and $b$ were the amplitude, peak orientation, standard deviation of the Gaussian function (equal to half width at half height), and minimal response, respectively.

The population orientation tuning curves for different eye preference groups were derived using the same procedure, with additional binning of neurons according to their ODI. To obtain the tuning curve of the neurons preferring the eye seeing the grating, responses of neurons with an ODI <–0.2 (preferentially responding to the contralateral eye) under contralateral eye grating presentation and those with an ODI >0.2 (preferentially responding to the ipsilateral eye) under ipsilateral eye grating presentation were combined. Similarly, for neurons preferring the eye seeing the masker, responses of neurons with an ODI <–0.2 under ipsilateral eye grating presentation and those with an ODI >0.2 under contralateral eye grating presentation were combined. For binocular neurons (–0.2<ODI<0.2), responses under both grating presentation conditions were combined.

## Fisher information

The Fisher information assesses the amount of information contained in a neuron population using an optimal decoder (**Pouget et al., 1999**). Assume independent Gaussian noise distributions, the Fisher information for a population of *N* neurons was given as

$$I = \sum_{i=1}^{N} \frac{f_i'\left(\theta\right)^2}{\sigma_i\left(\theta\right)^2}$$

where $f_i\left(\theta\right)$ was the mean activity of neuron *i* in response to the presentation angle, $\theta$, and $f_i'\left(\theta\right)$ was its derivative with respect to $\theta$. We fitted each neuron's response tuning $f_i\left(\theta\right)$ and variance tuning $\sigma_i\left(\theta\right)$ with Gaussian functions and calculated the averaged Fisher information across neurons at each orientation.

## SVM-based orientation classification

In the orientation classification task (**Figure 3A**), we trained an SVM with a one-vs.-one coding scheme to classify orientations from standardized population neural activity. The SVM decoder was implemented using MATLAB's '*fitcsvm*' function with a linear kernel. To prevent overfitting and evaluate the generalization ability of the model, we employed a 10 fold cross-validation procedure, and the model performance on the validation dataset was reported.

Decoders were trained independently for each experiment condition, resulting in four models per FOV (contralateral/ipsilateral × baseline/CFS). Neural response data from two spatial frequencies were used as input, with each neuron treated as a feature. In this way, each model was trained and tested on 336, 288 or 240 samples (2 SFs × 12 orientations × 14/12/10 repeats).

## The transformer model

### Model input, output, and training procedure

We implemented a transformer-based model to reconstruct noise-free grating stimuli from population neuronal responses recorded under different experiment conditions. The model input was a vector of neuronal responses, each corresponding to an individual neuron, and the output was the reconstructed grating image of size 70×70 pixels.

The transformer was trained independently for each experimental condition, resulting in four models per FOV (contralateral/ipsilateral × baseline/CFS). Pilot experiments revealed that our original dataset was insufficient for the model to converge. To address this, we augmented the dataset to four times its original size before training. Augmentation was performed by sampling from a normal distribution centered at each neuron's response mean, with a standard deviation equal to its original standard deviation. Within the augmented dataset, 6% was reserved for validation. Responses were normalized to [0, 1] before being fed into the model.

We implemented a two-phase training procedure to assess the reconstruction ability of models trained on different neural data. During the training process, the model typically reached two learning plateaus, where the validation loss temporarily stagnated (*Figure 3C*). In the first training session, we analyzed the learning curve to determine the epoch at which the baseline model's validation loss had completed 75% of its total decrease between the two plateaus. This was estimated using a modified sigmoid fit:

$$\text{loss}\left(t\right) = A - \frac{C}{1 + e^{k(b-t)}}$$

where $A$ and $C$ defined the function range, $b$ was the symmetry point, $k$ was the steepness parameter, and $t$ represented the epoch number, counted from epoch 500 (initial epochs were discarded due to a drastic drop in validation loss across all training runs, see *Figure 3C*). The 75% decrease point was computed as:

$$\text{Stop point} \ = \ b + \frac{\ln\left(3\right)}{k}$$

In the second training session, we retrained both models up to the identified epoch and evaluated their performances.

The model was trained to minimize the mean squared error (MSE) between the reconstructed and actual noise-free stimuli. Optimization was performed using RMSprop with a learning rate of 0.00005 and a smoothing factor $\rho$=0.85.

### Model structure

Each neuron's response was embedded into a higher-dimensional space using a learned weight vector as follows:

$$R^1 = W^{emb} \odot R^0$$

where the $R^0$ ($n$×1) represented the original response vector from $n$ neurons, and $W^{emb}$ ($n$×$d_{model}$) was the embedding weight matrix, with each row corresponding to a neuron-specific weight vector. Here, we used $d_{model}$=2. The symbol $\odot$ denoted row-wise multiplication, such that the $i$th response $r_i^0$ was multiplied by both elements in its embedding weight vector $w_i^{emb}$. The resulting embedding matrix $R^1$ ($n$×$d_{model}$) contained the high-dimensional representations of the neuronal responses.

The enriched embedding matrix was then passed through a self-attention module. In this module, $R^1$ was first projected into queries ($Q$), keys ($K$), and values ($V$) through independent learnable weight matrices, respectively. Then the attention map was computed as:

$$AttentionMap = SoftMax\left(\frac{QK^T}{\sqrt{d_k}}\right)$$

where $d_k$ represented the dimensionality of the key vectors, which scaled the dot product to control the variance of the attention scores.

The output of self-attention was calculated as:

$$R^2 = AttentionMap \times V$$

The output from self-attention was unembedded by projecting each neuron's high-dimensional representation back to one-dimensional. A feedforward layer transformed the unembedded vector into a stimulus vector, which was then reshaped into the final 70×70 image.

### Model evaluation

The original, non-augmented data was used for analysis, which had been seen during training in both the training and validation sets. We used an SSIM (**Brunet et al., 2012**) to quantify the reconstruction performances.

The SSIM (**Brunet et al., 2012**) between two images $x$ and $y$ (both 70×70) is defined as:

$$SSIM(x, y) = \frac{(2\mu_x\mu_y + c_1)(2\sigma_{xy} + c_2)}{(\mu_x^2 + \mu_y^2 + c_1)(\sigma_x^2 + \sigma_y^2 + c_2)}$$

where $\mu_x$ and $\mu_y$ are the mean intensities, $\sigma_x^2$ and $\sigma_y^2$ are the variances, $\sigma_{xy}$ is the covariance, and $c_1$ and $c_2$ are constants for numerical stability.

## Acknowledgements

This study was supported by a National Natural Science Foundation of China, Brain Science and Brain-like Intelligence Technology—National Science and Technology Major Project grant (2022ZD0204600) to SMT and CY. We thank Yi Jiang for commenting on an early version of this manuscript.

## Additional information

### Funding

| Funder | Grant reference number | Author |
|---|---|---|
| National Natural Science Foundation of China, Brain Science and Brain-like Intelligence Technology—National Science and Technology Major Project grant | 2022ZD0204600 | Cong Yu<br>Shi-Ming Tang |

The funders had no role in study design, data collection and interpretation, or the decision to submit the work for publication.

### Author contributions

Cai-Xia Chen, Data curation, Formal analysis, Visualization, Writing – original draft, Writing – review and editing; Xin Wang, Data curation, Formal analysis, Writing – review and editing; Dan-Qing Jiang, Data curation; Shi-Ming Tang, Resources, Software, Supervision, Funding acquisition, Methodology, Writing – review and editing; Cong Yu, Conceptualization, Funding acquisition, Writing – original draft, Writing – review and editing

### Author ORCIDs

Cai-Xia Chen ⓘ https://orcid.org/0009-0004-3140-0426
Xin Wang ⓘ https://orcid.org/0009-0004-3016-0547
Shi-Ming Tang ⓘ https://orcid.org/0000-0003-0294-3259
Cong Yu ⓘ https://orcid.org/0000-0002-8453-6974

### Ethics

All experimental protocols were approved by the Peking University Animal Care and Use Committee (LSC-TangSM-3). All surgery was performed under general anesthesia and strictly sterile condition, and every effort was made to minimize suffering.

Reviewer #1 (Public review): https://doi.org/10.7554/eLife.107518.4.sa1
Reviewer #2 (Public review): https://doi.org/10.7554/eLife.107518.4.sa2
Reviewer #3 (Public review): https://doi.org/10.7554/eLife.107518.4.sa3
Author response https://doi.org/10.7554/eLife.107518.4.sa4

## Additional files

### Supplementary files
MDAR checklist

### Data availability
The code can be found at GitHub: https://github.com/caviaryusi/CFS_2p (copy archived at *Chen, 2026*). The data can be found at Zenodo (Zenodo, RRID:SCR_004129): https://doi.org/10.5281/zenodo.20053907.

The following dataset was generated:

| Author(s) | Year | Dataset title | Dataset URL | Database and Identifier |
| --- | --- | --- | --- | --- |
| Chen C, Wang X, Jiang D-Q, Shi-Ming T, Yu C | 2026 | Data for "Continuous flashing suppression of neural responses and population orientation coding in macaque V1" | https://doi.org/10.5281/zenodo.20053907 | Zenodo, 10.5281/zenodo.20053907 |

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
