## [Editor Report · eLife Assessment]

This **important** study shows that orientation tuning of V1 neurons is suppressed during a continuous flash suppression paradigm, especially in neurons with binocular receptive fields. These findings, made using cutting-edge imaging techniques, **convincingly** implicate early visual processing in continuous flash suppression, in agreement with previous studies suggesting reduced effective contrast of such stimuli in V1.

---

## [Referee Report · Reviewer #1 (Public review)]

[Editors' note: this version has been assessed by the Reviewing Editor without further input from the original reviewers. The authors have submitted a second revision, largely to address a comment from Reviewer 2, which was "The failure to model the neural data with an explicit model is a missed opportunity." The authors have now included a computational model.]

This study makes a fundamental contribution to our understanding of interocular suppression, particularly continuous flash suppression (CFS). Using neuroimaging data from two macaque monkeys, the study provides compelling evidence that CFS suppresses orientation responses in neurons within V1. These findings enrich the CFS literature by demonstrating that neural activity under CFS may prevent high-level visual and cognitive processing.

Comments on previous revisions:

The authors have addressed all my previous comments.

---

## [Referee Report · Reviewer #2 (Public review)]

Summary:

The goal of this study was to investigate the degree to which low-level stimulus features (i.e., grating orientation) are processed in V1 when stimuli are not consciously perceived under conditions of continuous flash suppression (CFS). The authors measured the activity of a population of V1 neurons at single neuron resolution in awake fixating monkeys while they viewed dichoptic stimuli that consisted of an oriented grating presented to one eye and a noise stimulus to the other eye. Under such conditions, the mask stimulus can prevent conscious perception of the grating stimulus. By measuring the activity of neurons (with Ca2+ imaging) that preferred one or the other eye, the authors tested the degree of orientation processing that occurs during CFS.

Strengths:

The greatest strength of this study is the spatial resolution of the measurement and the ability to quantify stimulus representations during CSF in populations of neurons preferring the eye stimulated by either the grating or the mask. There have been a number of prominent fMRI studies of CFS, but all of them have had the limitation of pooling responses across neurons preferring either eye, effectively measuring the summed response across ocular dominance columns. The ability to isolate separate populations offers an exciting opportunity to study the precise neural mechanisms that give rise to CFS, and potentially provide insights into nonconscious stimulus processing.

Weaknesses:

(The authors have now included a computational model in the second revision.)

---

## [Referee Report · Reviewer #3 (Public review)]

Summary:

In this study, Tang, Yu & colleagues investigate the impact of continuous flash suppression (CFS) on the responses of V1 neurons using 2-photon calcium imaging. The report that CFS substantially suppressed V1 orientation responses. This suppression happens in a graded fashion depending on the binocular preference of the neuron: neurons preferring the eye that was presented with the marker stimuli were most suppressed, while the neurons preferring the eye to which the grating stimuli were presented were least suppressed. Binocular neuron exhibited an intermediate level of suppression.

Strengths:

The imaging techniques are cutting-edge.

Weaknesses:

The strength of CFS suppression varies across animals, but the authors attribute this to comparable heterogeneity in the human psychophysics literature.

Comments on previous revisions:

The authors have addressed my comments from the previous round of review, and I have no further comments.

---

## [Author Response]

The following is the authors’ response to the previous reviews

**Public Reviews:**

**Reviewer #1 (Public review):**
This study makes a fundamental contribution to our understanding of interocular suppression, particularly continuous flash suppression (CFS). Using neuroimaging data from two macaque monkeys, the study provides compelling evidence that CFS suppresses orientation responses in neurons within V1. These findings enrich the CFS literature by demonstrating that neural activity under CFS may prevent high-level visual and cognitive processing.Comments on revisions:The authors have addressed all my previous comments.

Thanks for the very warm comments!

**Reviewer #2 (Public review):**
Summary:The goal of this study was to investigate the degree to which low-level stimulus features (i.e., grating orientation) are processed in V1 when stimuli are not consciously perceived under conditions of continuous flash suppression (CFS). The authors measured the activity of a population of V1 neurons at single neuron resolution in awake fixating monkeys while they viewed dichoptic stimuli that consisted of an oriented grating presented to one eye and a noise stimulus to the other eye. Under such conditions, the mask stimulus can prevent conscious perception of the grating stimulus. By measuring the activity of neurons (with Ca2+ imaging) that preferred one or the other eye, the authors tested the degree of orientation processing that occurs during CFS.Strengths:The greatest strength of this study is the spatial resolution of the measurement and the ability to quantify stimulus representations during CSF in populations of neurons preferring the eye stimulated by either the grating or the mask. There have been a number of prominent fMRI studies of CFS, but all of them have had the limitation of pooling responses across neurons preferring either eye, effectively measuring the summed response across ocular dominance columns. The ability to isolate separate populations offers an exciting opportunity to study the precise neural mechanisms that give rise to CFS, and potentially provide insights into nonconscious stimulus processing.Weaknesses:However, while this is an impressive experimental setup, the major weakness of this study is that the experiments don't advance any theoretical account of why CFS occurs or what CFS implies for conscious visual perception. There are two broad camps of thinking with regard to CFS. On the one hand, Watanabe et al., 2011 reported that V1 activity remained intact duringCFS, implying that CFS interrupts stimulus processing downstream of V1. On the other hand, Yuval-Greenberg and Heeger (2013) showed that V1 activity is in fact reduced during CFS. By using a parametric experimental design, they measured the impact of the mask on the stimulus response as a function of contrast, and concluded that the mask reduces the gain of neural responses to the grating stimulus. They presented a theoretical model in which the mask effectively reduced the SNR of the grating, making it invisible in the same way that reducing contrast makes a stimulus invisible.In the first submission of the manuscript, the authors incorrectly described the Yuval-Greenberg & Heeger (2013) paper and Watanabe et al. (2011) papers, suggesting that they had observed the same or similar effects of CFS on V1 activity, when in fact they had described opposite results. Reviewer 1 also observed that the authors appeared to be confused in their reading of these highly relevant papers. In the revision, the authors have reworked this paragraph, now correctly describing these sets of opposing results. However, I still do not understand what the authors are trying to argue: "...these studies were not designed to quantify the pure effect of CFS on stimulus-evoked V1 responses." I do not understand what is meant by "pure" in this case.

This is clarified as: “Nevertheless, these studies contrasted monocular and dichoptic masking conditions to equate stimulus input while manipulating perceptual visibility, which were not designed to quantify the pure effect of CFS on stimulus-evoked V1 responses, that is, the difference of BOLD signals between binocular masking and stimulus alone conditions.” (line 63)

Regardless, it is clear that the measurements in the present study strongly support the interpretation of Yuval-Greenberg & Heeger (i.e., that V1 activity is degraded by CFS, 'akin' to a loss in the contrast-to-noise ratio of neural activity). It would be appropriate for the authors to communicate this clearly.

We agree and added the following sentence in the text: “These results support the conclusion of Yuval-Greenberg and Heeger (2013) that V1 activity is degraded by CFS, ‘akin’ to a loss in the contrast-to-noise ratio of neural activity” (line 122)

I continue to be of the opinion that this study is lacking an adequate model of interocular interactions that might explain the Ca2+ imaging. The machine learning results are not terribly surprising - multivariate methods, such as SVMs, are more sensitive than univariate approaches. So it is plausible that an SVM can support decoding of the coarse orientation information, even when no tuning is evident in the univariate analyses. However, the link between this result and the underlying neurophysiology is opaque. The failure to model the neural data with an explicit model is a missed opportunity.

We agree and put “An ocular-dominance-dependent gain control model” back to the text. Fig. 2D now shows the results of model fitting.

(line 167)

An ocular-dominance-dependent gain control model

We developed an ocular dominance-dependent gain control model to account for the impact of CFS on V1 population orientation tuning. The model development followed two steps.

Step I. Population orientation tuning functions before CFS

The population orientation tuning functions due to monocular stimulation exhibited different amplitudes among OD groups (Fig. 2D, red curves), which could be simulated with Equation 1, an OD-weighted Gaussian basis function:\begin{document}$$\displaystyle \boldsymbol{R}=\boldsymbol{w}^{s} \times\left[A \times \exp ^{-\left(\frac{\theta}{\sigma}\right)^{2}}+B\right]$$\end{document}

where parameters *A*, *σ*, and *B* corresponded to the amplitude, standard deviation, and minimal response of the Gaussian basis function, respectively, and θ represented the preferred orientation of a bin of neurons relative to the actual orientation of the grating stimulus. The weight parameter *w* was the mean of linearly transformed ODIs of neurons in a neuronal group, which equated to *(ODI +1)/2* or *1 - (ODI + 1)/2*, depending on contralateral or ipsilateral eye grating stimulation, and ranged from 0-1. Thus, a smaller *w* would indicate a higher preference for the eye seeing the grating, and a larger *w* would indicate a higher preference for the unstimulated eye (or the eye seeing the flashing masker under CFS). The *w* equated to 0.33, 0.50, and 0.67 in Monkey A, and 0.32, 0.5, and 0.68 in Monkey B, for the grating eye-preferring group, binocular group, and the masker eye-preferring group, respectively. The exponent *s* represented a nonlinear transformation.

Equation 1 fitted the baseline data well (Fig. 2D, red curves), resulting in goodness-of-fit (*R2*) values at 0.94 and 0.95 for the two monkeys, respectively. This indicated that the equation captured the OD-dependent population orientation tuning characteristics of V1 neurons with monocular stimulation before CFS.

Step II. The impacts of CFS

In step II, the model introduced several binocular combination factors to account for population orientation tuning functions under CFS.

To account for the OD-dependent changes of orientation tuning bandwidths under CFS, a *w*-dependent inhibition factor *wt* was introduced, which scaled the σ of the tuning functions, changing the monocular tunings R into R’:\begin{document}$$\displaystyle \mathbf{R}^{\prime}=\boldsymbol{w}^{s} \times\left[\mathrm{A} \times \exp ^{-\left(\frac{\theta}{\sigma \times \boldsymbol{w}^{t}}\right)^{2}}+\mathrm{B}\right]$$\end{document}

This allowed different groups of neurons to exhibit various degrees of orientation tuning function broadening, capturing the pattern in which neurons preferring the eye seeing the grating displayed a sharper population orientation tuning curve under CFS than those preferring the eye seeing the masker.

Previous studies have shown that binocular neuronal responses can be modeled by incorporating interocular suppression and summation processes (Kato et al., 1981; Dougherty, Cox, Westerberg, & Maier, 2019; Zhang et al., 2024). Therefore, R’ was further normalized by the neural response to the flashing masker to simulate interocular suppression, which was the first component of Equation 3. Additionally, the neural response to the flashing masker was summed to simulate binocular summation, which was the second component of Equation 3. These two components when summed, determining the final neural responses under CFS:\begin{document}$$\displaystyle \mathbf{R}_{C F S}=\frac{R^{\prime}}{a N^{k}}+b N^{m}$$\end{document}

where *N* was the empirical neural response to the monocularly presented flashing masker stimulation, *a* and *b* were scaling parameters, and *k* and *m* were nonlinearity parameters. The interocular normalization by masker response led to amplitude reduction of population orientation tuning functions for different groups of neurons, while the binocular summation with masker response elevated the minimal responses of tuning functions to their corresponding heights.

During the step II model fitting, the parameters *A*, *σ*, and *s* were inherited from the monocular tuning fits derived from Equation 1 and served as inputs, while the parameters *a*, *k*, *b*, *m*, and *t* were optimized. The model captured the CFS modulation on population orientation tuning curves well, with *R2* = 0.99 and 0.98 for Monkeys A and B, respectively (Fig. 2D, red curves).

**Reviewer #3 (Public review):**
Summary:In this study, Tang, Yu & colleagues investigate the impact of continuous flash suppression (CFS) on the responses of V1 neurons using 2-photon calcium imaging. The report that CFS substantially suppressed V1 orientation responses. This suppression happens in a graded fashion depending on the binocular preference of the neuron: neurons preferring the eye that was presented with the marker stimuli were most suppressed, while the neurons preferring the eye to which the grating stimuli were presented were least suppressed. Binocular neuron exhibited an intermediate level of suppression.Strengths:The imaging techniques are cutting-edge.Weaknesses:The strength of CFS suppression varies across animals, but the authors attribute this to comparable heterogeneity in the human psychophysics literature.Comments on revisions:The authors have addressed my comments from the previous round of review, and I have no further comments

Thanks!